# Assessment of Innovative Dry Powders for Inhalation of a Synergistic Combination Against *Mycobacterium tuberculosis* in Infected Macrophages and Mice

**DOI:** 10.3390/pharmaceutics17060705

**Published:** 2025-05-27

**Authors:** Faustine Ravon, Emilie Berns, Isaline Lambert, Céline Rens, Pierre-Yves Adnet, Mehdi Kiass, Véronique Megalizzi, Cédric Delporte, Alain Baulard, Vanessa Mathys, Samira Boarbi, Nathalie Wauthoz, Véronique Fontaine

**Affiliations:** 1Unit of Microbiology, Bioorganic and Macromolecular Chemistry, Faculty of Pharmacy, Université Libre de Bruxelles (ULB), 1050 Brussels, Belgiumisaline.lambert@ulb.be (I.L.); veronique.megalizzi@ulb.be (V.M.); 2Unit of Pharmaceutics and Biopharmaceutics, Faculty of Pharmacy, Université Libre de Bruxelles (ULB), 1050 Brussels, Belgium; nathalie.wauthoz@ulb.be; 3Unit “Tuberculosis & Mycobacteria”, Human Bacterial Diseases, Sciensano, 1180 Brussels, Belgiummehdi.kiass@sciensano.be (M.K.); vanessa.mathys@sciensano.be (V.M.); samira.boarbi@sciensano.be (S.B.); 4Unit of Pharmacognosy, Bioanalysis and Drug Discovery & Analytical Platform of the Faculty of Pharmacy (APFP), Faculty of Pharmacy, Université Libre de Bruxelles (ULB), 1070 Brussels, Belgium; cedric.delporte@ulb.be; 5Centre d’Infection et d’Immunité de Lille, Institut Pasteur de Lille, Université de Lille, CNRS, Inserm, U1019-UMR9017, 59000 Lille, France; alain.baulard@pasteur-lille.fr

**Keywords:** pulmonary delivery, tuberculosis, vancomycin, orlistat, macrophage infection model

## Abstract

**Background/Objectives:** In vitro, vancomycin (VAN) and tetrahydrolipstatin (THL) together have been shown to synergistically inhibit *Mycobacterium tuberculosis* (Mtb), the world’s most infectious killer. The poor oral bioavailability of VAN and THL and predominant tropism of Mtb infection to the lungs and alveolar macrophages make pulmonary administration highly attractive. This study aimed to develop and assess the efficacy of dry powders for inhalation of VAN microparticles embedded with THL. **Methods**: The dry powders produced by spray-drying, with or without hydrogenated castor oil (HCO), were characterized for their physicochemical properties among others by HPLC-DAD. The fast-screening impactor was used to determine powder aerodynamic properties, and VAN and THL releases were established from the paddle over disk method. Biological activities were assessed in a new *M. bovis*-infected macrophage model and in Mtb-infected mice. **Results and Discussion**: The addition of 25% HCO enables co-deposition (fine particle dose) at the desired weight ratio and co-releasing of VAN and THL in aqueous media. Microparticles with 0% to 50% HCO drastically reduced cytoplasmic *Mycobacterium bovis* survival (99.9% to 62.5%, respectively), with higher efficacy at low HCO concentration. Consequently, VAN/THL with or without 25% HCO was evaluated in Mtb-infected mice. Although no decrease in Mtb lung burden was observed after two weeks of administration, the endotracheal administration of VAN 500 mg/kg and THL 50 mg/kg with 25% HCO administrated three times during five days concomitantly with daily oral rifampicin (10 mg/kg) demonstrated 2-fold bacterial burden reduction compared to the group treated with RIF alone. **Conclusions**: HCO was crucial for obtaining a fine particle dose at the synergistic weight ratio (VAN/THL 10:1) and for releasing both drugs in aqueous media. With oral administration of the first-line rifampicin, the dry powder VAN/THL/25% HCO was able to exert a potential anti-tubercular effect in vivo in Mtb-infected mice after five days.

## 1. Introduction

Tuberculosis (TB) is estimated to cause 1.25 million deaths in the world due to a single infectious agent, *Mycobacterium tuberculosis* (Mtb) [1]. Treating TB remains a global challenge. The latest World Health Organization (WHO) guidelines for the treatment of drug-susceptible TB still recommend a 6-month rifampicin-based oral regimen (2 months of isoniazid, rifampicin (RIF), pyrazinamide and ethambutol, followed by 4 months of isoniazid and RIF) [1]. This prolonged multidrug oral regimen, with its numerous systemic side effects, adversely affects patient compliance, adherence and persistence to medication, leading to treatment failure, relapse of infection, and the emergence of multidrug-resistant TB [1]. Additionally, this systemic oral treatment can result in drug–drug interactions that may reduce the efficacy of other systemically administered medications.

The causative agent of TB, *Mycobacterium tuberculosis*, primarily resides in the lungs. Delivering TB treatment via the pulmonary route is a promising strategy, though it has yet to yield a marketed drug. The pulmonary route (or inhalation), like oral routes recommended by WHO, is a non-invasive route that increases patients’ compliance and persistence to medication, by reducing the need for hospital visits for treatments. Moreover, inhalation allows for deposition of high drug concentration directly at the site of infection, minimizing exposure to uninfected tissues, thereby limiting systemic side effects and preventing drug–drug interactions often encountered with polymedication [2,3,4,5]. These high concentrations administered directly to the lungs minimize systemic side effects and help prevent multidrug resistance [2,3,4,5]. Dry powder inhalers (DPIs) and nebulizers are the two types of inhalation devices used to deliver high drug dosage to the lungs. For infectious diseases, DPIs were chosen over nebulizers. Dry forms are more stable over time and better suited for drugs with varying physicochemical properties such as water solubility [6]. DPIs are portable and less expensive than nebulizers and allow for shorter administration time (i.e., 1 to 5 min compared to 15 min for nebulizers) [6]. Additionally, the aerosol through a DPI is generated by the patient’s inhalation, drastically reducing environmental exposure compared to nebulizers, where the aerosol is continuously generated and only a small fraction of the aerosol is inhaled and deposited in the lungs (i.e., 10–15%) [6]. Consequently, they require no external power supply (patient’s inspiratory flow versus electricity) and less maintenance post-treatment. Therefore, DPIs reduce the risk of exposing other patients or healthcare workers to Mtb or antibiotics during and after treatment [6].

Alveolar macrophages (AMs) are the primary host cells for Mtb in pulmonary TB. Once phagocytosed, Mtb escapes the phagolysosome and resides in the cytosol, utilizing AM resources for their growth [7]. Therefore, to be effective, anti-TB drugs must first be deposited into the lungs, particularly in the alveoli, to contact AM, reach the AM cytoplasm, and ultimately cross the highly impermeable, lipid-rich cell envelope of Mtb [2,3,4,5,8]. To reach the AM cytoplasm, two strategies can be envisaged: either the drugs diffuse through the AM cell membrane to act on cytosolic Mtb, or the microparticles are internalized by phagocytosis and escape or diffuse through the phagolysosome membrane to reach the AM cytoplasm [3].

Therefore, anti-TB drugs need to meet specific characteristics to reach intracellular Mtb. Firstly, the aerodynamic diameter (d_ae_) of the drug-based formulation should be between 1 and 5 µm, with a high proportion between 1 and 3 µm for optimal alveolar deposition and AM phagocytosis [9,10]. The goal is to achieve a fine particle fraction (FPFn), (i.e. the fraction in percentage related to the nominal dose that presents a d_ae_ < 5 µm) higher than 35%, and an emitted fraction (i.e., the fraction in percentage related to the nominal dose emitted through the DPI) higher than 90%. These criteria are essential for good dispersion and/or aerosolization properties of the dry powder through the chosen DPI. Additionally, the deposited particles must remain sufficiently undissolved in lung fluid to allow phagocytosis by AM.. A size between 1 and 3 µm, spherical shape, hydrophobic surface, and excipients that slow down the particle dissolution rate are crucial factors to enhance AM phagocytosis [10,11,12,13,14]. Maximum phagocytosis has been observed for particles in the range of 1–2 μm, within 1 h incubation [10].

Vancomycin hydrochloride (VAN), a large, freely water-soluble, and hygroscopic tricyclic antibiotic, targets peptidoglycan synthesis and is widely used to treat infections caused by Gram-positive bacteria. However, in the presence of Mtb cell wall lipid biosynthesis inhibitors, such as THL, which alter mycobacterial lipid metabolism including phthiocerol dimycocerosate (PDIM) production, this antibiotic has been able to target Mtb [15,16]. Typically, THL, also called orlistat, is a small and practically insoluble drug in water, prescribed orally to combat obesity by inhibiting pancreatic lipases. As an anti-TB drug candidate, THL increases the permeability of the Mtb cell wall, allowing the relatively large VAN to diffuse into Mtb and exert its pharmacological action. The combination of VAN and THL has shown an in vitro synergistic effect at 3:1 *w*/*w* (10 µg/mL and 3.1 µg/mL) and 10:1 *w*/*w* (10 µg/mL and 1 µg/mL, respectively) [16].

In terms of tolerance, inhaled VAN has demonstrated safety in numerous studies [17,18,19], including those involving humans [18,19]. When combined with THL, no interaction or toxicity increase was demonstrated [17]. However, THL exhibits reversible cytostatic effects on lung cells in vitro, but only at high concentrations (IC50 observed at 50–57 µg/mL for THL, a concentration around 50 times its concentration within the VAN/THL combination, inhibiting more than 99% of the Mtb population (MIC_99_)) [16,17]. At 100 times the MIC_99_ (i.e., THL 100 µg/mL with or without 1000 µg/mL VAN), no significant irreversible decrease in the transepithelial electrical resistance was observed on a Calu-3-based monocellular epithelium [17]. The in vivo tolerance studies, performed three times a week for three weeks (subacute exposure) with VAN/THL in solution at 50 times the MIC_99_ (i.e., 500 mg/kg VAN and 50 mg/kg THL), showed no in vivo lung toxicities in terms of inflammation, cytotoxicity, and other histopathological processes [17]. A weight ratio of 10:1 between VAN and THL was chosen (rather than 3:1) for further pharmaceutical development, THL being the potential limiting factor in the combination in terms of tolerance.

The aim of this work was to develop dry powders for inhalation where VAN and THL were in a weight ratio of 10:1 with aerodynamic properties allowing them to be co-deposited in this proportion in the lung and subsequently to be phagocytosed by AM. Once phagocyted, the dry powders should co-release VAN and THL to ensure their synergistic pharmacological action against Mtb.

Two different formulation strategies were developed: the first one involved embedding/coating the hygroscopic VAN microparticles with the practically insoluble THL, and the second one involved embedding/coating VAN microparticles with THL and different proportions of hydrogenated castor oil (HCO) (i.e., 25 and 50%). The hydrophobic microparticle surface could indeed counteract VAN hygroscopic properties, thereby reducing water absorption, which was essential to improve microparticle dispersion properties by decreasing capillary forces, thereby reducing the dissolution rate of microparticles, and increasing phagocytosis by AM with minimal use of excipients.

Additional aspects were also considered. Indeed, considering the dilution effect caused by the high surface area of alveoli of approximatively 100–150 m^2^ and the risk of Mtb-resistant strains, concentrations of at least 10 times higher than the MIC_99_ need to be delivered to the alveoli. Inversely, the lung fluid volume being quite low (i.e., 10–20 mL), this would favor higher drug concentration in the lungs. Furthermore, the dry particles needed to be partly phagocytized by AM, while the other part needed to be gradually dissolved in the lung fluid or lung parenchyma to target extracellular Mtb.

HCO was chosen as the lipid in the formulation due to its high melting point (i.e., 85 °C), making it compatible with the spray-drying process, and its approval for oral delivery [20,21]. Additionally, HCO has demonstrated safety in pulmonary administration during regulatory toxicity evaluations in rats and dogs, supporting its use in clinical phases for humans [22]. Furthermore, HCO has been shown to decrease the release of cisplatin in the dry formulation, while the PEGylated derivative of tocopherol succinate (TPGS) increases lung retention by limiting AM phagocytosis [22,23,24]. In this study, HCO was used without TPGS to decrease the microparticle dissolution and enhance their phagocytosis by AM.

## 2. Materials and Methods

The material is described in detail in Appendix A.

### 2.1. Production of Dry Powders for Inhalation Based on VAN and THL

#### 2.1.1. First Step of Production: Micronization of VAN

The size reduction of VAN was initially prepared at laboratory scale, by spray-drying (SD-VAN). VAN was dissolved in water at a solid content of 1% (*w/v*). The solution was then spray-dried using a Mini Spray Dryer B-290 (Büchi Labortechnik AG, Flawil, Switzerland). The following parameters were used during the process: feed rate, 2.4 mL/min; maximum drying air flow, 35 m^3^/h; nozzle size, 0.7 µm; maximum compressed air, 819 L/min; inlet temperature, 130 °C, leading to an outlet temperature of 70 ± 2 °C. This inlet temperature was used to evaporate water from a solution containing the hygroscopic VAN and to reduce the moisture content of the dried particles without VAN degradation. The resultant powder was blown through the cyclone separator and collected in a pharmaceutical-grade glass vial, sealed with rubber stoppers and aluminum caps. The vials were then stored in a closed receptacle with desiccant under vacuum at room temperature.

#### 2.1.2. Second Step of Production: Preparation of Dry Powder for Inhalation

Firstly, THL was dissolved in a mixture of methyl acetate/ethanol absolute (1:3 *v/v*) under magnetic stirring. If necessary, different proportions of HCO (0%, 25%, or 50%) were added to the heated solution (45–50 °C). Then, SD-VAN was dispersed in this solution. The final concentration of solid content (i.e., VAN, THL, and/or HCO) was 1% *w/v*. The suspension was then spray-dried with the following parameters: feed rate, 2.4 mL/min; maximum drying air flow, 35 m^3^/h; nozzle size, 0.7 µm; maximum compressed air, 819 L/min; inlet temperature, 60 °C; leading to an outlet temperature of 39 ± 1 °C. This inlet temperature was used to evaporate both class 3 organic solvents without softening THL which has a melting point of 44 °C. The resultant powder was blown through the cyclone separator and collected in a pharmaceutical-grade glass vial, sealed with rubber stoppers and aluminum caps. The vials were stored in a closed receptacle with desiccant under vacuum at room temperature.

### 2.2. In Vitro Characterization of the SD-VAN and Dry Powders for Inhalation

#### 2.2.1. Physicochemical Properties

VAN and THL determination—HPLC-DAD method

The analytical determination of VAN and THL in the samples was determined using a High-Performance Liquid System (HPLC) connected to a diode array detector (DAD) (Agilent Technologies, Santa Clara, CA, USA). The system was equipped with a binary pump and an autosampler. A 4.6 × 150 mm Symmetry HPLC C18 column from Waters with a particle diameter of 5 µm was used for separation. A gradient elution system was optimized (0 to 100% Solution B) at a flow rate of 1 mL/min with a mobile phase consisting of a mixture of 2 solutions: Solution A was 20 mM NH_4_ formic buffer pH 3.2 adjusted with formic acid and Solution B was acetonitrile/Solution A (90:10 *w/w*). Detection was performed with a DAD system at 281 and 203 nm and an injection volume of 5 µL and 40 µL for VAN and THL, respectively. The total running time was 27 min. All the HPLC analyses were performed at room temperature. The method was firstly validated before using it for drug content determination (Appendix A, Appendix A, Appendix A).

VAN and THL content

VAN and THL contents (%) in the dry powders for inhalation were assessed on a minimum of 3 samples of accurately weighted 20.00 mg dissolved in a volumetric flask containing the dilution phase (i.e., mixture of ultrapure water and absolute ethanol (75:25 *v/v*)). Samples were sonicated and heated at 35 °C for 30 min. The lipid solutions were filtered with a 0.2 µm polycarbonate filter before analytical determination by the validated HPLC-DAD method (Appendix A, Appendix A).

Residual solvent, HCO content and yield

The residual solvent content (%) was evaluated using thermogravimetric analysis (TGA) using a Q500 thermogravimetric analyzer (TA Instruments, New Castle, DE, USA) and Universal Analysis 2000 software version 4.5A (TA Instruments, New Castle, DE, USA). Runs were performed in triplicate on 5–7 mg sample, set with platinum pan from 30 to 200 °C at a heating rate of 10 °C/min. Weight losses were assessed between 35 and 105 °C to determine the total residual solvent content using Universal Analysis 2000 software v. 4.5A.

HCO content (%) was calculated by subtracting the residual solvent, THL, and VAN contents. Yield (%) is the percentage of the solid content recovered after the spray-drying in comparison with the solid content before the spray-drying.

#### 2.2.2. In Vitro Aerodynamic Properties

The fast-screening impactor (FSI) from Copley Scientific Ltd. (Nottingham, UK) was used to determine the fine particle dose (FPD) (or fine particle fraction, FPF) of the dry powders for inhalation. FSI separates particles lower than 5 µm of d_ae_ on a 90 mm glass fiber filter (1 µm pore size). The quantity of VAN and THL deposited at each stage of the device (capsule, DPI, induction port adapter) and FSI (induction port, pre-separator, and filter) was recovered using the dilution phase (ultrapure water and absolute ethanol (75:25 *v/v*)) and determined using the validated HPLC-DAD method (Appendix A, Appendix A).

For each dry powders for inhalation, an accurate mass of approximatively 20.00 mg was placed in a size 3 HPMC capsule after being sieved through a 315 µm stainless steel mesh. The filled capsule was inserted into a RS01 DPI with low resistance (Plastiape S.p.A., Osnago, Italy). The flow rate was controlled using a DFM 2000 Flowmeter (Copley Scientific Ltd., Nottingham, UK) associated with two high-capacity pumps model HCP5 and coupled to a critical flow controller model TPK 2000 (Copley Scientific Ltd., Nottingham, UK). It was applied at 100 ± 5 L/min for 2.4 s, with a critical flow with a P3/P2 ratio lower than 0.5 to allow a depression of 4 Kpa through the DPI as recommended by European Pharmacopeia for aerodynamic determination of fine particles.

Each assay was performed using 10 capsules, as 10 is the maximum number recommended by European Pharmacopeia. The HPLC-DAD method was used to determine the VAN or THL content deposited on each stage.

#### 2.2.3. In Vitro Release Properties

The release properties of VAN and THL from the FPF of the dry powders for inhalation were established using a previously described in vitro dissolution method derived from the paddle over the disk [25]. The watch glass/patch/PTFE assembly was submerged in a dissolution vessel of a 708-DS dissolution apparatus (Agilent Technologies, Machelen, Belgium). The dissolution medium was 500 mL phosphate-buffer saline (PBS) containing 3% of sodium dodecyl sulphate (SDS) in the aim to discriminate formulations by detecting both drugs while guaranteeing sink conditions. The temperature was maintained at 37 ± 0.5 °C. Paddles were set at 25 ± 2 mm between the blade and the center of the assembly, with a speed set at 50 rpm. Sample aliquots were collected from 5 min to 24 h and replaced with the same volume of pre-heated dissolution medium. Samples were then filtered with a 0.7 µm pore size polycarbonate filter before determining the amount of VAN and THL using the validated HPLC-DAD method (Appendix A, Appendix A). To establish the 100% drug release, the watch glass/patch/PTFE was disassembled, and the filter, the mesh, and the clips were placed into the dissolution vessels with the paddle set at 200 rpm for 2 h. Dissolution assays were carried out in triplicate for each formulation. For dissolution assay, fit factors f_1_ and f_2_ were calculated. The fit factors f_1_ (difference factor) and f_2_ (similarity factor) were used to determine the similarity of dissolution profiles of formulations [26], according to VAN and THL release. If f_1_ is smaller than 15 (close to 0) and f_2_ is higher than 50 (close to 100), dissolution/release profiles are similar. Otherwise, dissolution/release profiles are considered different.

### 2.3. In Vitro Efficacy Study on Infected Macrophages

#### 2.3.1. Monocyte Differentiation into Macrophages

The THP-1 human monocytic leukemic cell line was cultured in RPMI medium supplemented with 10% fetal bovine serum at 37 °C and 5% CO_2_. THP-1 differentiation in macrophage was conducted 72 h before infection in a 24-well plate with 2.5 × 10^5^ cells/mL and 50 ng/mL phorbol 12-myristate 13 acetate (PMA). The plates were then incubated for 72 h at 37 °C. Before infection, the PMA medium was removed and cells were washed three times with 1 mL of PBS.

#### 2.3.2. Bacterial Culture and Infection of Macrophages

The *Mycobacterium bovis* BCG strain (BCG::ESX-1^Mmar^), provided by Roland Brosch from the Institut Pasteur (Paris, France), produced the *Mycobacterium marinum* ESX-1 secretion system [27]. BCG::ESX-1^Mmar^ was grown in 7H9 medium supplemented with 10% ADC and 0.05% Tween 80 at 37 °C under a BSL-2 laboratory.

Bacterial cultures were used when OD_600nm_ was between 0.6 and 0.8. Prior to infection, mycobacteria were washed 3 times with PBS before being mixed in RPMI supplemented with 10% FBS. Clumped mycobacteria were pelleted by centrifugation at 4000× *g* for 5 min, and homogeneous supernatants were used for infection. BCG::ESX-1^Mmar^ bacterial suspension was concentrated at 2.5 × 10^6^ colony forming units (CFU)/mL and used to replace the culture medium of macrophages (multiplicity of infection of 10). Phagocytosis was allowed to occur for 4 h at 37 °C. Macrophages were then washed 3 times with PBS before being further treated with amikacin (200 µg/mL for 2 h at 37 °C) to kill extracellular mycobacteria.

#### 2.3.3. Treatment and Analysis

After phagocytosis and amikacin treatment, infected macrophages were washed 3 times with PBS and then exposed to treatment for 24 h or 4 days at 37 °C. The dry powders for inhalation were weighed and diluted in the RPMI medium at a concentration of 500 µg/mL VAN and 50 µg/mL THL, corresponding to 50 times the combination MIC_99_ [16]. Controls were performed: a positive one to control the infection (infected macrophages not treated) and a negative one to control the sterility of the assays (macrophages not infected and not treated).

A total of 24 hours or 4 days after treatment, macrophages were washed 3 times with PBS and lysed with 200 µL cell-lysis buffer (1% of SDS dissolved in PBS). Mycobacteria were collected in the cell-lysis buffer by scrapping the culture dish. The suspension was then subjected to vigorous shaking for 30 s. The number of intracellular live mycobacteria was assessed by serial dilution of the lysate. A total of 20 microliters was plated on agar plate (solid medium 7H11 supplemented with 0.5% glycerol and 10% OADC) for determination of CFU/mL by colony counting after 15 days of incubation at 37 °C. To evaluate the efficacy of the formulation against mycobacteria, the percent reduction was calculated using the following formula:$$Percent\;reduction=\frac{(A-B)\times 100}{A}$$
where *A* corresponds to the number of bacteria viable without treatment, and *B* corresponds to the number of bacteria after treatment.

### 2.4. In Vivo Efficacy Study in Mtb-Infected Mice

The animal protocol is registered under ethical protocol BioEthic Number 20200107-01 (registration number LA1230177, approval date 10 March 2020) obtained by the Ethical Commission of Sciensano (Brussels, Belgium). All experiments were performed in a BSL3.

#### 2.4.1. Animals

Male (approximately 22.8 g) and female (approximately 18.6 g) BALB/cByJ mice, aged 6 weeks, purchased from Charles River (Saint Germain Nuelles, France), were used after a quarantine period. The animal had free access to food and water. Mice were kept in controlled standard conditions (day/night cycle of 12 h, temperature of 22 ± 2 °C, humidity rate of 55 ± 10%). A maximum of 5 animals of the same gender were placed in filtertop cage Type III H (Carfil Quality, Oud-Turnhout, Belgium).

Body weights were recorded 3 times per week. The animals were examined once per day for appearance, posture, and behavior. During the study, humane limit endpoints were defined by using a scoring. When the mice reach the maximal humane endpoint scoring, the euthanasia was performed as requested.

#### 2.4.2. Infection

The H37Rv Lux *M. tuberculosis* strain (producing PDIM and luciferase) [28] was grown on 7H9 medium supplemented with 10% OADC until exponential phase was reached (OD_600nm_ = 0.6). Mycobacteria were washed to obtain concentrated and homogeneous preparation. The washes were carried out by centrifuging the culture at 4000× *g* for 15 min. The supernatant was removed, and the pellet was resuspended with PBS. The culture was then syringed using a 27G needle to break up any remaining clumps of bacteria and glycerol was added (25% final concentration). Before infection, the bacterial load was determined by plating various inoculum dilutions on 7H11 agar plates supplemented with oleic-albumin-dextrose complex (OADC). The number of CFU/mL was counted after 5–6 weeks. The inoculum was aliquoted and stored at −80 °C until use.

After the quarantine period, mice were anesthetized using a solution of ketamine (150 mg/kg) and xylazine (2 mg/kg) intraperitoneally and infected with the H37Rv Lux Mtb strain at a dose of 1 × 10^5^ per mouse by delivering a 50 µL volume to the back of the throat using a micropipette. Treatment began one week post-infection.

#### 2.4.3. Preparation and Characterization of the Powder Blends for Pulmonary Administration in Mice

Since dry powders for humans cannot be emitted uniformly using the Powder Administration Device for Animals (PADA) (Aptar Pharma, Crystal Lake, IL, USA), an excipient was used as a diluent and to improve the emitted dose uniformity [29]. Therefore, the dry powders (VAN/THL with 0% or 25% HCO) for in vivo treatments were blended with Lactohale^®^ 300 as previously made for other spray-dried dry powders [24]. All the dry powders for inhalation and diluents were sieved through a 315 µm stainless steel mesh and blended in a 10 mL glass. Blends were mixed using a three-dimensional Powder Blender Mixer Turbula^®^ (Willy A. Bachofen AG, Muttenz, Switzerland) with a rotation speed of 50 rpm twice for 15 min with a sieving step in between.

The in vitro emission efficiency in mass and in dose of the blends through PADA are described in Appendix A, Appendix A.

#### 2.4.4. Treatment

One week after infection, treatments were administered, with anesthesia performed as described in Section 2.4.2. The powder delivery to the lung was performed using the powder blends described in Section 2.4.3 by the endotracheal route using the PADA, 3 times a week (Monday, Wednesday, and Friday) for 2 weeks. Groups of mice (equal quantity of males and females) were treated by the blends (B-VAN/THL/25% HCO or B-VAN/THL) to deliver 500 mg/kg of VAN and 50 mg/kg THL or by blend made with spray-dried HCO as a vehicle. The 8 mice from the control group of infected mice (4 males and 4 females) did not receive any treatment or anesthesia. The number of mice per group and duration of treatment are described in Appendix A (Appendix A).

#### 2.4.5. Sampling, Histopathology, and Determination of the Relative Light Unit (RLU)

Mice were euthanized by cervical dislocation. Lungs were collected and grinded in 5 mL of a mixture of 2.5% of MB/BacT antibiotic supplement in PBS, using a gentleMACS™ Dissociator (Miltenyi Biotec, Bergisch Gladbach, Germany). Relative light units (RLU) were quantified for each grinded lung in duplicate. An amount of 15 µL ZAP-OGLOBIN II Lytic Reagent was added to 1 mL grinded lung, and the suspension mixture was centrifugated for 5 min at 4000× *g*. The supernatant was thrown and 900 µL of PBS was added. An amount of 100 microliters of 1% decanal (in ethanol) was added. Measurement of RLU was performed using a LB 9508 Lumat^3^ tube luminometer (Berthold Technologies, Bad Wildbad, Germany). The software version 1.0.4 was used to analyze data.

### 2.5. In Vitro and In Vivo Efficacy Studies of VAN/THL Dry Powder for Inhalation Associated with First-Line Anti-TB Drug RIF

In vitro efficacy study in Mtb

The H37Rv Mtb strain was grown in 7H9 medium containing 0.05% polysorbate 80 supplemented with 10% ADC. Drug susceptibility was investigated using the standardized agar proportion method [30]. A H37Rv Mtb strain suspension inoculum was used (1 MacFarland turbidity) diluted from 10^−1^ to 10^−4^ to spot the different dilution of the initial culture on a 24-well plate containing ½ diluted drug series in 7H11 agar plates. THL and RIF were serially diluted alone or in combination with a fixed concentration of RIF (0.01 µg/mL) or a fixed concentration of THL (1 µg/mL), respectively.

The MIC of THL or RIF alone or in combination was used to investigate synergy by the Checkboard method [31]. A fractional inhibitory concentration index (FICI) was calculated as follows:$$FICI={FIC}_{a}+{FIC}_{b}=\frac{{MIC}_{ab}}{{MIC}_{a}}+\frac{{MIC}_{ba}}{{MIC}_{b}}$$

Synergy was reached when *FICI* ≤ 0.5, indifference was reached when 0.5 < *FICI* < 4, and an antagonistic effect was reached when *FICI* > 4 [32].

In vivo efficacy study in Mtb-infected mice

RIF powder was suspended in a dispersion of 0.5% carboxymethylcellulose (CMC) and 0.5% polysorbate 80 in water. The mixture was vortexed and stirred for 30 min at 50 °C. The suspension was then poured into glass vials at 10 or 7.5 mg/kg and stored at 4 °C until use. RIF was applied by gavage via a canula (100 µL per mouse). One week after infection, treatment was applied.

The first experiment was planned for 2 weeks but lasted 1 week due to excessive weight loss until the humane endpoint (20% of initial weight). This experiment uses a group of 14 mice (7 males and 7 females) that was endotracheally treated by B-VAN/THL/25% HCO at VAN 500 mg/kg and THL 50 mg/kg using the PADA, 3 times a week (Monday, Wednesday, and Friday), after ketamine/xylazine anesthesia and by oral RIF at 10 mg/kg by gavage, 5 times a week (Monday up to Friday). Mice were treated by RIF at 10 mg/kg, 5 times a week (Monday up to Friday) and were used as a control. The number of mice per group and duration of treatment are described in Appendix A (Appendix A).

As humane endpoints were reached, further investigations were carried out before the second experiment to assess the reasons for weight loss (Appendix A, Appendix A).

The second experiment was planned for 2 weeks and lasted 2 weeks without reaching humane endpoints, thanks to the supplementation of Solid Drink^®^ allowing for overcoming of the negative cumulative effect of oral and endotracheal administrations on body weight. A group of mice was treated by B-VAN/THL at VAN 500 mg/kg and THL 50 mg/kg using the PADA, 3 times a week under anesthesia and by oral RIF that was decreased at 7.5 mg/kg for more security, by gavage 5 times a week. The numbers of mice in the endotracheal-treated groups were reduced as no death was encountered during the endotracheal administrations under anesthesia during the previous experiment. Mice were treated by RIF at 7.5 mg/kg by gavage 5 times a week during 1 week and were used as a control. Euthanasia, sampling, and RLU quantification were performed in the same way as that described in Section 2.4.5. The number of mice per group and duration of treatment are described in Appendix A.

### 2.6. Statistical Analysis

The number of animals in each group was based on the results obtained by the software G power using a priori evaluation. Alpha and beta parameters were fixed at 0.05 and 0.80, respectively. GraphPad Prism^®^ software (Version 8.4.3) was used to perform statistical analyses. A t-test was applied for comparing 2 groups. An ANOVA and Bonferroni’s multiple comparisons tests were applied when comparing more than 2 groups. The statistical results were expressed as follows: non-significant for *p* > 0.05, statistically significant (*) for *p <* 0.05, very significant (**) for *p <* 0.01, and extremely significant for *p <* 0.001 (***) and for *p <* 0.0001 (****). 

## 3. Results and Discussion

### 3.1. Production of Dry Powders for Inhalation and Physicochemical Properties

The size of raw VAN was firstly reduced by spray-drying (SD-VAN) to obtain particles with a particle size distribution (PSD) between 1 and 5 µm. After the spray-drying process, the median particle size (Dv(50)) of VAN decreased from 76 ± 4 µm to 3.0 ± 0.2 µm (Appendix A, Appendix A). This result demonstrated that 50% of VAN particles after the spray-drying process were below 3 µm, an optimal particle size to reach the alveoli after pulmonary administration [9,10]. The span, representing the PSD width, was 1.5 for SD-VAN, indicating a homogeneous and unimodal PSD, as span is inferior to 2 [9]. Moreover, the Dv(90) was 5.58 ± 0.09 µm for SD-VAN, meaning that 90% of microparticles were below 5.58 µm (Appendix A, Appendix A). This is the upper size limit to reach the lungs after inhalation if the particle density is close to 1 and the particle shape is close to a sphere. Next, we aimed to suspend SD-VAN in a heated solution containing dissolved THL and different percentages of HCO (0%, 25%, or 50% *w/w*) to embed/coat VAN microparticles within a THL or THL/HCO matrix. The contents of VAN and THL, moisture/solvent levels, and calculated HCO in the spray-dried formulations are presented in Table 1 (details are presented in Appendix A).

Determination of drug content in the dry powders for inhalation demonstrated that the 10:1 weight ratio was maintained after the spray-drying process for the dry powders for inhalation, except for VAN/THL without HCO, where a decrease in VAN was observed (Table 1).

Moisture content on the microparticle surface is a crucial factor as it influences the adhesion and/or cohesion between microparticles. The absence of moisture content can lead to electrostatic forces on the microparticle surface, which may impair dispersion properties. Moisture can neutralize electrostatic charges and thus be beneficial for dispersion properties leading to higher FPD and FPF. However, beyond a certain level, moisture can increase capillary forces between microparticles, which impair dispersion properties [29,33,34]. As VAN is a hydrophile and hygroscopic molecule [35], the level of 8.13 ± 0.02% after spray-drying was logical. The thermogram profile corresponds to adsorbed water, as observed by a progressive loss of water between 35 and 105 °C, whereas bound water is only removed between 100 and 125 °C (Appendix A).

As VAN/THL and VAN/THL/25–50% HCO were formulated in a mixture of methyl acetate and absolute ethanol (both class 3 solvents), a low amount of residual solvent was required to reduce the risk on health. Class 3 residual solvents are indeed limited to not more than 0.5% per day [36]. A TGA analysis was performed on spray-dried formulations containing only THL or THL combined with HCO (formulated under the same conditions as described in Section 2.1.2) and no residual solvent was found (0 ± 0%). It would therefore appear that the residual solvent in VAN/THL/0–50% HCO dry powders was solely adsorbed water supplied by SD-VAN due to its hygroscopic properties.

The addition of a practically insoluble molecule such as THL at 10% with or without 25% or 50% of HCO progressively decreased the moisture content by dilution effect, as shown in Table 1 and Appendix A.

The dry powders for inhalation consisted of microparticles with a doughnut-shaped spherical morphology and a smooth surface (Appendix A). The PSD of all dry powders for inhalation was characterized by a homogeneous and unimodal (span below 2) PSD with a Dv(10) of 1.3 µm, Dv50 of 3.0 µm, and a Dv90 of 5.0 µm (Appendix A, Appendix A), which is quite favorable for reaching the lungs and alveoli after inhalation.

In terms of crystalline properties, raw VAN and SD-VAN exhibited an amorphous content of 100% according to XRPD analysis (Appendix A, Appendix A), indicating no crystalline components, which aligns with the absence of a melting point observed during DSC analysis for both raw and SD-VAN (Appendix A, Appendix A). Raw THL and raw HCO were predominantly characterized by specific crystalline peaks in XRPD, confirmed by their endothermic melting points observed during DSC analysis (Appendix A, Appendix A), i.e., 44 °C for THL [37] and between 85 and 88 °C for HCO, which is a mixture of lipids [38]. Spray-dried HCO from a solvent solution presents a crystalline state of the most stable β polymorph [24]. The dry powder for inhalation VAN/THL is characterized by a predominantly amorphous state due to the amorphous VAN, with a portion of THL in a crystalline state, as confirmed by the endothermic melting point observed at 44 °C during DSC analysis (Appendix A, Appendix A) and a decrease in the amorphous content from 100% to 97% during XRPD analysis, along with specific THL crystalline peaks (Appendix A, Appendix A). The addition of HCO to the dry powders for inhalation decreased the amorphous content from 97% to 93% and 89% for 25% and 50% HCO (Appendix A, Appendix A), respectively, as confirmed by the endothermic melting peak ranging from 61 to 86 °C and specific β polymorph HCO crystalline peaks (Appendix A, Appendix A).

The melting point corresponding to the THL in VAN/THL dry powder disappeared in the DSC thermograms of dry powders for inhalation with 25% and 50% HCO Appendix A), suggesting that THL is completely solubilized in the β polymorph HCO matrix (Appendix A).

The amorphous/crystalline content and moisture content in SD formulations remains stable over time even after 9 months (Appendix A, Appendix A), which is beneficial in terms of long-term stability. Moisture can facilitate the crystallization of amorphous content, leading to a decrease in both amorphous content and moisture content [29]; however, this is not the case here.

### 3.2. In Vitro Aerodynamic Performances of the Dry Powders for Inhalation

The deposition in % of THL and VAN from the different dry powders for inhalation in the different parts of the device and FSI is presented in Figure 1.

The total dose recovered for each formulation ranged from 91 to 100% for VAN and 79–106% for THL of the total nominal dose, ensuring the validity of the FSI determination as recommended by the European Pharmacopoeia for the aerodynamic determination of fine particle dose using an impactor (i.e., 85–115%). The most important parameter was the amount of powder deposited on the filter, corresponding to the FPD (i.e., the dose of drug-based particles having a d_ae_ < 5 µm) or FPFn (i.e., the fraction of FPD of the nominal dose). This represents the inhalable dose or fraction capable of reaching the lungs.

For 200 mg of dry powder (i.e., 10 capsules filled with 20 mg), the highest FPD for VAN was obtained with VAN/THL at 51.2 mg, followed by VAN/THL/25% HCO at 41.9 mg, followed by VAN/THL/50% HCO at 31.7 mg, primarily due to the dilution effect of the HCO matrix. In fact, the FPF% was not significantly different (*p ˃* 0.05) among the three formulations, with all around 30%.

For THL, the highest FPD was obtained with VAN/THL HCO50% at 4.4 mg, followed by VAN/THL/25% HCO at 4.03 mg, and finally with VAN/THL at 1.21 mg. The HCO matrix drastically and significantly decreased the amount of THL in the pre-separator and drastically and significantly increased the amount into the filter, corresponding to the amount reaching the lungs (FPF and FPD). This effect overcame the HCO dilution effect, as there was a significant difference between VAN/THL and VAN/THL/25–50% HCO (*p* ˂ 0.001 and *p* ˂ 0.0001, respectively).

To achieve the synergistic effect, a weight ratio between VAN and THL ranging from 3:1 to 10:1 is recommended [16]. The weight ratio obtained in the filter and therefore in the lung was 42:1 for VAN/THL, 10:1 for VAN/THL/25% HCO and 7:1 for VAN/THL/50% HCO. Therefore, VAN/THL/25% HCO and VAN/THL/50% HCO were both the best formulations for depositing the synergistic weight ratio of VAN and THL into the lung (3:1 to 10:1).

After deposition and AM phagocytosis, microparticles need to release VAN and THL with similar kinetics to exert their synergistic effects. Therefore, dissolution tests were conducted from the inhalable fraction deposited on the FSI filter (i.e., for microparticles with a d_ae_ < 5 µm).

To compare the dissolution release profile, the difference f_1_ and similarity f_2_ factors were calculated. VAN and THL from an inhalable fraction of VAN/THL dry powder exhibited different release profile over (f_1_ above 15 and f_2_ below 50), demonstrating a faster dissolution of THL compared to VAN (Figure 2). For VAN/THL/25% HCO and VAN/THL/50% HCO dry powders, the release profiles of VAN and THL were not different (f_1_ below 15 and f_2_ above 50) (Figure 2). Thus, in the presence of HCO, the drugs were co-dissolved, allowing for potential synergistic action.

It should be noted that the dissolution medium used in this work was not physiological (i.e., presence of 3% SDS in PBS) but was necessary to allow determination of both drugs by HPLC-DAD method. Therefore, the presence of 3% SDS increased the dissolution rate of THL and VAN from the microparticles compared to physiological conditions encountered in the lungs. THL is more difficult to solubilize under physiological conditions and slows down the release of VAN before its complete dissolution. This could potentially favor AM phagocytosis of the undissolved parts of the formulation, while the dissolved parts could act either by diffusing through the AM cell membrane or by acting on extracellular Mtb. The developed spray-dried formulations demonstrated favorable characteristics for pulmonary administration. The VAN/THL/25% HCO dry powder appears to be the best candidate, combining a microparticle size suitable for lung delivery in appropriate weight and for co-release of both drugs.

### 3.3. In Vitro Efficacy Study in M. bovis BCG::ESX-1^Mmar^-Infected Macrophages

The in vitro efficiency of the different dry powders for inhalation (VAN/THL, VAN/THL/25% HCO and VAN/THL/50% HCO) and the excipient (HCO) was assessed in a newly developed macrophage infection model, using the *M. bovis* BCG::ESX-1^Mmar^ strain able to be maintained in the macrophage cytosol. Indeed, this strain expresses the ESX-1 secretion system, which plays a key role in host–pathogen interactions [39]. Although the ESX-1 system is not present in *M. bovis* BCG, it is present in *Mycobacterium marinum* and in the recombinant BCG::ESX-1^Mmar^ expressing the ESX-1 from *M. marinum* [27]. Therefore, this in vitro efficacy assay allows for the assessment of drug efficiency against cytosolic mycobacteria. A 24 h in vitro dry powder for inhalation treatment exposure was performed. The number of intracellular surviving bacteria was assessed by CFU counting from cell lysates of infected macrophages, and the percent reduction of BCG::ESX-1^Mmar^ viability was calculated and compared with untreated macrophages (Ctrl). As shown in Figure 3A,B, excipient HCO alone had no statistically significant effect on the bacterial burden after 24 h and 4 days (*p* > 0.05, Appendix A).

A statistically significant decrease in CFU/mL retrieved from infected macrophages was observed after 24 h and 4 days for all tested dry powders for inhalation, including VAN/THL/50%, 25% or 0% HCO at a concentration of 500 µg/mL of VAN and 50 µg/mL of THL (50 times the combination MIC_99_). The percent reduction of BCG::ESX-1^Mmar^ viability after 24 h treatment was between 27.9% for the VAN/THL without HCO, 45.9% for the VAN/THL/50% HCO, and 52.5% VAN/THL/25% HCO. VAN/THL dry powder for inhalation without HCO demonstrated the strongest in vitro antimycobacterial activity against BCG::ESX-1^Mmar^ compared to VAN/THL/25% HCO and VAN/THL/50% HCO (respectively: *p* = 0.1848; *p <* 0.0001). The higher amount of HCO in VAN/THL/50% HCO led to a significant lower decrease in BCG::ESX-1^Mmar^ viability (*p* = 0.0004).

To improve the detection of antimycobacterial activity, the same assay was performed for 4 days (Figure 3B). The three dry powders for inhalation statistically decreased BCG::ESX-1^Mmar^ viability compared to the negative control. The reduction ranged from 69.2 to 99.9%. VAN/THL dry powder without HCO demonstrated the best in vitro inhibitory action (99.9% decrease) against *M. bovis* BCG::ESX-1^Mmar^ compared to VAN/THL/25% HCO 25% (83.2%) and VAN/THL/50% HCO (69.2%) (*p <* 0.0001, ****). Moreover, VAN/THL and VAN/THL/25% HCO were significantly more efficient than VAN/THL/50% HCO (*p <* 0.0001, ****) confirming the results obtained after 24 h (Appendix A).

Nevertheless, the *M. bovis* BCG::ESX-1^Mmar^ bioburden was only decreased in the presence of VAN/THL dry powders with 10-fold higher concentrations than the MIC (Figure 3 shows results obtained with 50-fold higher concentrations than the MIC). 

Based on these results, VAN/THL and VAN/THL/25% HCO dry powders for inhalation were selected for in vivo efficiency assay in Mtb-infected mice.

### 3.4. In Vivo Efficacy Studies of the Dry Powder for Inhalation VAN/THL and VAN/THL/25% HCO in Mtb-Infected Mice

Blends of dry powders for inhalation designed for endotracheal administration using the PADA insufflator were initially characterized to ensure an appropriate and reproducible emitted dose. The in vitro emission efficiency in dosage through the PADA was determined in ten doses (Appendix A, Appendix A). The in vitro efficiency in dose presented a CV% around 10% for both formulations. This variation is highly acceptable, as the “Uniformity of delivered dose of preparations for inhalation” specified in the European Pharmacopeia v.10 [40] requires all 10 emitted doses to fall between 65 and 135% of the mean, with at least 9 doses between 75 and 125%.

Due to spontaneous mutations of PDIM biosynthesis genes, Mtb is inclined to lose its ability to produce PDIM, but its presence is necessary to ensure optimal virulence of Mtb via its impermeable cell membrane. The loss of PDIM would therefore lead to poor experimental results. Before performing animal experiments, the presence of PDIM in the inoculum was verified.

The in vivo efficacy study was conducted on mice infected with the H37Rv Lux Mtb strain, allowing for the assessment of Mtb burden in lung extracts by measuring luciferase activity (in RLU/mL) [41].

Unfortunately, VAN/THL/25% HCO and VAN/THL-based blends did not reduce the Mtb burden in the infected preclinical murine model of pulmonary TB over two weeks (Figure 4A,B). A group of mice was treated with HCO alone to verify that the excipient did not influence the growth of Mtb. No statistical action of HCO was demonstrated in this assay (*p ˃* 0.05) (Appendix A).

Despite the promising in vitro results (Figure 3), the lack of efficacy of the dry powders for inhalation in this in vivo assay was unexpected (Figure 4). Several factors could explain this outcome. Firstly, the in vitro and in vivo studies were conducted on different strains, BCG::ESX-1^Mmar^ and Mtb H37RvLux, respectively. Secondly, pharmacodynamic and pharmacokinetic parameters could influence in vivo drug efficiency and should be further characterized in the future.

Indeed, the frequency of treatment can be questioned. It may be that the spray-dried formulations were eliminated too quickly and/or that the frequency of three times a week was not enough. This preclinical model could also be improved. Indeed, BALB/c mice develop cellular granulomas with minimal necrosis on TB infection, whereas necrotizing, caseous lesions are a hallmark of human TB. These differences influence local drug effects. Moreover, the tolerance of mice to Mtb infection may underestimate new treatments, as the host does not display a strong inflammatory response against the bacilli. For example, the use of guinea pigs could be a valuable alternative animal model, as it is considered the closest model to the human TB pathological process [42,43,44]. Nevertheless, the in vitro results have encouraged us to continue exploring these dry powders for inhalation in a different way, in combination with one of the main drugs used in the first-line oral treatment of TB, RIF, which similarly to vancomycin can act in synergy with THL.

### 3.5. Evaluation of the Dry Powders for Inhalation in Combination with Rifampicin

#### 3.5.1. In Vitro Efficiency of RIF and THL Against Mtb

We previously reported that an Mtb strain without PDIM in its cell membrane could be five times more susceptible to RIF than wild-type Mtb [15]. Since THL can reduce PDIM A and B level in Mtb [16], we investigated whether THL could synergize with RIF. This was studied using the standardized agar proportion method. The first step was to calculate the MIC of the THL alone, of the RIF alone, and of the combination of these two drugs (Table 2). Synergy between THL and RIF was assessed by calculating the FICI [33].

The MIC_99_ of THL alone was 5 to 10 µg/mL on the H37Rv Mtb strain, as previously reported [16]. RIF alone presented an MIC_99_ of 0.25 to 1 µg/mL against Mtb. In combination with 0.01 µg/mL of RIF, the MIC_99_ of THL felt to 0.3 to 1.25 µg/mL. The FICI calculated with the Checkboard method was between 0.185 and 0.3, indicating synergy between THL and RIF (Table 2).

#### 3.5.2. In Vivo Efficacy Studies of the Dry Powders for Inhalation in Combination with Oral RIF in Mtb-Infected Mice

An in vivo efficacy study was conducted to test efficacy of the blends based on VAN/THL/25% HCO (500 mg/kg VAN and 50 mg/kg THL), with or without oral RIF (10 mg/kg), on infected mice.

During the first experiment, an unexpected weight loss until the humane endpoint (20% from the initial weight) was observed in the mice group receiving VAN/THL/25% HCO and oral RIF. This was attributed to the stress generated by the accumulation of administration techniques (i.e., gavage and endotracheal administration with moderate duration of general anesthesia), leading to reduced food and water intake as demonstrated by further investigation conducted after this experiment (Appendix A, Appendix A, Appendix A). This group (VAN/THL/25% HCO and oral RIF) reached humane endpoints and was therefore sacrificed after one week. The oral RIF control group was also sacrificed to evaluate if the VAN/THL/25% HCO had any beneficial effect against pulmonary TB. Compared to oral RIF alone, the blend based on VAN/THL/25% HCO demonstrated a statistically significant decrease in Mtb burden after one week of treatment (*p* = 0.0110) (Figure 5A).

According to previous in vitro characterization and efficacy assay results, VAN/THL/25% HCO allows appropriate co-deposition and co-release. Although the previous in vivo assay did not demonstrate any inhibitory activity against Mtb when administered without RIF for 2 consecutive weeks, its activity against Mtb seemed potentiated in association with RIF over a shorter treatment time. As TB is always treated with a combination of drugs to avoid drug resistance, this finding could be valuable for potentially shortening drug regimen treatment and improving bactericidal activity.

A second experiment was conducted using the blend based on VAN/THL (without HCO in view of the results obtained on infected macrophages) with oral RIF at a lower dose (7.5 mg/kg rather than 10 mg/kg) and with Solid Drink^®^ supplementation, following the results obtained from the in vivo tolerance (Appendix A, Appendix A). The 2-week treatment administration of endotracheal B-VAN/THL associated with oral RIF had no statistically significant effect compared to RIF alone (*p* = 0.0879) (Figure 5B). Additionally, no toxicity issues were observed, confirming the positive role of Solid Drink^®^ supplementation in mitigating the impact of the accumulation of administration techniques.

According to in vitro dissolution and aerodynamic characterization, the hypothesis is that VAN/THL dry powder cannot act synergistically on Mtb because VAN and THL might not be co-released without HCO (Figure 2) especially in a 10:1 *w/w* ratio.

## 4. Conclusions

The present study demonstrated the feasibility of formulating dry powders of VAN and THL for pulmonary administration. Although various VAN/THL (HCO) inhibited *M. bovis* BCG::ESX-1^Mmar^ in infected macrophages, in vivo, only the formulation with 25% HCO inhibited Mtb in infected mice and only in the presence of oral rifampicin, one of the four first-line oral drugs used against Mtb.

Indeed, HCO is a crucial excipient in the dry powder to ensure co-release of VAN and THL from the microparticles and to deposit the appropriate VAN/THL weight ratio in the lungs.

## Figures and Tables

**Figure 1 pharmaceutics-17-00705-f001:**
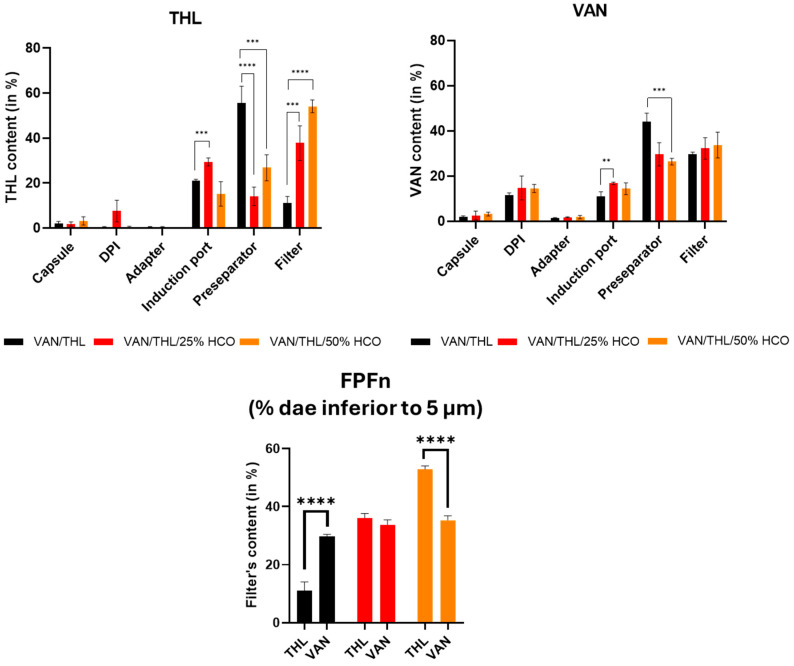
THL (on left) and VAN (on right) deposition (in % related to the nominal dose in the capsules using 10 capsules filled with 20 mg of dry powder) in the different parts of the device (capsule, DPI, adapter) and FSI (induction port, pre-separator, and filter). A focus was made about VAN and THL deposition in the filter representing the fine particle fraction (d_ae_ inferior to 5 µm) related to nominal dose. All results are expressed as means of VAN or THL content ± SD (*n* = 3). Results were very significant for *p <* 0.01 (**); extremely significant for *p <* 0.001 (***) and extremely significant with *p <* 0.0001 (****).

**Figure 2 pharmaceutics-17-00705-f002:**
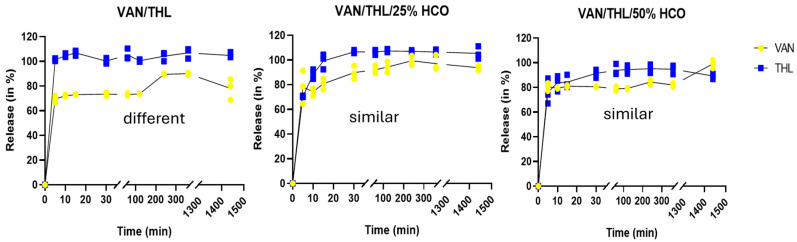
Release of VAN and THL at 37 ± 0.5 °C from inhalable particles of dry powders for inhalation impacted on the FSI filter and recovered with a filter (i.e., the assembly), in a dissolution medium of 500 mL phosphate-buffer saline (PBS) containing 3% of sodium dodecyl sulfate (SDS). All results are expressed as mean percentage of VAN or THL release ± SD (*n* = 3).

**Figure 3 pharmaceutics-17-00705-f003:**
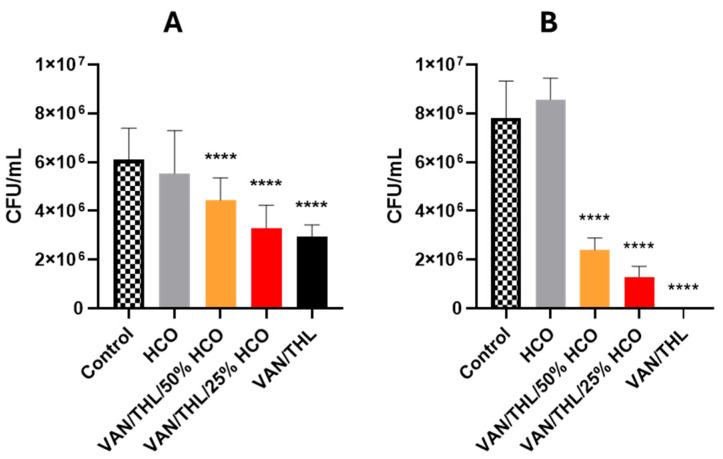
Impact of dry powder treatments on bacterial burden in macrophages infected with the BCG::ESX-1^Mmar^ strain. Treatment was performed for (**A**) 24 h or (**B**) 4 days. All results are expressed as means of CFU/mL ± SD (*n* = 3). The statistical analyses were performed versus the negative control group (extremely significant for *p <* 0.0001 (****)).

**Figure 4 pharmaceutics-17-00705-f004:**
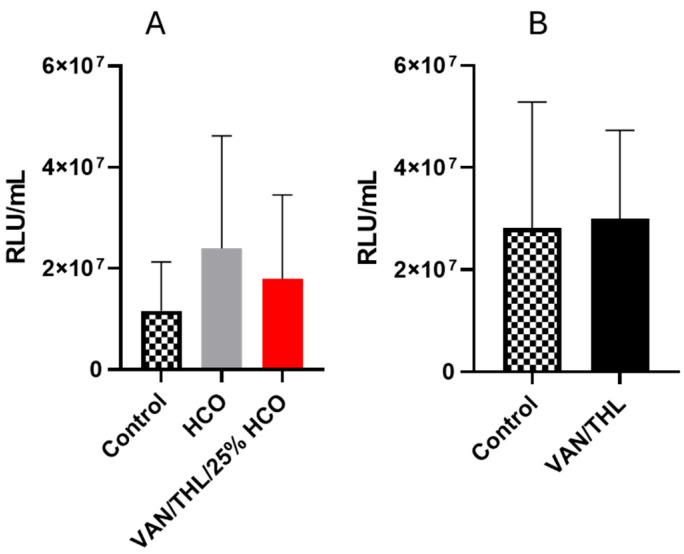
In vivo efficiency assay of blends based on (**A**) VAN/THL/25% HCO (VAN 500 mg/kg and THL 50 mg/kg) (*n* = 20) or HCO (*n* = 16) and (**B**) VAN/THL (VAN 500 mg/kg and THL 50 mg/kg) (*n* = 5), administered by endotracheal route under anesthesia 3 times a week during 2 consecutive weeks. All results are expressed as means of RLU/mL ± SD (*n* = 5–20). The *p*-values are higher than 0.05 (non-significant), in comparison to untreated infected mice (CRT = 14 and 8, for **A** and **B**) (Appendix A).

**Figure 5 pharmaceutics-17-00705-f005:**
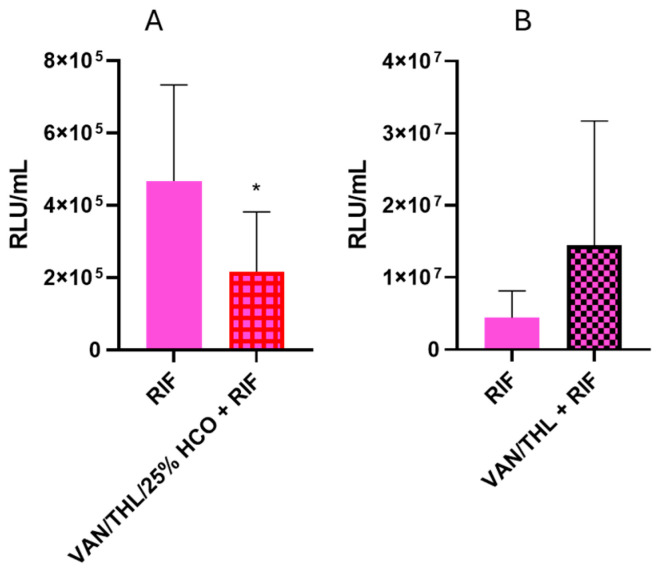
In vivo efficacy assay of blends based on (**A**) VAN/THL/25% HCO (VAN 500 mg/kg and THL 50 mg/kg) three times a week in association with oral RIF at 10 mg/kg five times a week after 1 week of treatment without nutritive supplementation (*n* = 12) and (**B**) VAN/THL (VAN 500 mg/kg and THL 50 mg/kg) three times a week in association with oral RIF at 7.5 mg/kg five times a week after 2 weeks of treatment with nutritive supplementation (*n* = 10). All results are expressed as means of RLU/mL ± SD (*n* = 10–14). The statistical analyses were performed versus the oral RIF control group, at 10 mg/kg (*n* = 12), panel A, or at 7.5 mg/kg (*n* = 10), panel B (significant for *p <* 0.05 (*)).

**Table 1 pharmaceutics-17-00705-t001:** Content (%) of VAN and THL (weight ratio) determined by the validated HPLC-DAD method, moisture determined by TGA, HCO calculated by subtracting VAN, THL and moisture content, and yield (%) calculated by ratio of the recovered mass after spray-drying to the initial weighted mass before spray-drying.

Formulations	VAN Content (%)	THL Content (%)	Residual Solvent (%)	HCO Content (%)	Yield (%)
SD-VAN	97 ± 1	NA	8.13 ± 0.02	NA	85
VAN/THL	86 ± 6 (7.8)	11 ± 2 (1)	7.95 ± 0.02	NA	76
VAN/THL/HCO 25%	65 ± 2 (10.1)	6.4 ± 0.4 (1)	5.39 ± 0.04	23.1	70
VAN/THL/HCO 50%	47 ± 2 (10.4)	4.8 ± 0.4 (1)	4.71 ± 0.02	43.5	69

Results are expressed as means ± SD (*n* = 3), except Yield and HCO content. NA: not applicable.

**Table 2 pharmaceutics-17-00705-t002:** Synergy between THL and RIF to inhibit H37Rv Mtb based on the fractional inhibitory concentration index calculation.

Drugs	MIC_99_ (µg/mL)	FICI
RIF	0.25–1	0.185–0.3
RIF (+THL 1 µg/mL)	<0.06–1.25	
THL	5–10	
THL (+RIF 0.01 µg/mL)	0.3–1.25	

All results are expressed as means of MIC in µg/mL ± SD (*n* = 3).

## Data Availability

Data supporting reported results can be found online from DOI: 10.5281/zenodo.15007137.

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
