# Peer review of "Assessment of Innovative Dry Powders for Inhalation of a Synergistic Combination Against Mycobacterium tuberculosis in Infected Macrophages and Mice"

_pharmaceutics, 2025, doi:10.3390/pharmaceutics17060705_

Round 1
Reviewer 1 Report
Comments and Suggestions for Authors
Dear Authors
After a thorough revision, I have the following queries and suggestions to improve the quality of this manuscript.
• Were the spray-drying process parameters used for preparing SD VAN previously optimized or referenced in earlier studies? Please clarify.
• What is the residual moisture content in the final SD VAN product? Why was the standard KF titration not used to determine it? The TGA cannot distinguish between water, organic solvents and hydrates, and would require thermal events very well separated? I could not also see the TGA thermograms in the supplementary data.
• What is the solubility of vancomycin HCl (VAN HCl) in ethanol? If it is indeed soluble in ethanol (verify), the use of the term “dispersed” may not be appropriate—please justify this terminology.
• Why did the authors not consider using a solvent system or emulsion system capable of dissolving both THL and VAN HCl to improve homogeneity and product quality?
• Can you explain the reasons why doughnut shaped particles were formed?
• Please provide references for the in vivo infection model used.
• What was the rationale for blending the final spray-dried product with Lactohale? Were the aerodynamic properties of the neat spray-dried formulation inadequate for aerosolization?
• Were the aerosolization parameters (e.g., MMAD, GSD, fine particle fraction) for the spray dried and Lactohale blend measured and reported? If not, this data should be included.
• Was the number of animals in each treatment and control group determined using statistical power analysis? If so, please justify the parameters used; if not, explain how the group sizes were determined.
• Why was an intratracheal infection model used instead of a more physiologically relevant low-dose whole-body aerosol exposure model for pulmonary TB?
• What is the rationale for dosing animals with the DPI formulation three times per week compared to the standard five-day oral rifampicin regimen?
• Is THL thermally stable after its first-order thermal transition at 44 °C? Please discuss the implications of this transition on the spray-drying process and the stability of the final formulation.
• Several figure images are blurry. Please provide higher-resolution versions.
• Can the authors compute and provide detailed aerosolization parameters such as MMAD, geometric standard deviation (GSD), and respirable fraction? This information is essential and should be included in the results.
• From the drug release studies, it appears that 80–95% of VAN and THL are released within 10 minutes. Is this release window sufficient for uptake by alveolar macrophages? Are the particles sufficiently solid and stable for intracellular targeting to infected macrophages or the lung interstitium?
• The statistical analysis in Figure 3A should be carefully rechecked. The error bars, especially for the VAN/THL/50% HCO group, do not seem to support the claimed statistical significance.
• Why were different strains of Mycobacterium used for the in vitro and in vivo studies? This discrepancy affects the translational value of the data. The authors may consider either presenting in vitro data using H37Rv or omitting M. bovis data altogether to ensure consistency.
• Why was RFU (Relative Fluorescence Units) chosen as the primary readout for bacterial burden instead of the standard CFU plating method? RFU data tends to be highly variable and may not reliably distinguish between bacteriostatic and bactericidal effects.
• The manuscript’s readability could be improved by moving non-essential to the supplementary section and streamlining the Results and Discussion sections or more appropriately as separate “Discussions” section. The results are too mixed and there is not a proper story. The appendix section makes it more confusing.
Author Response
Dear reviewer,
Please open the attachment to obtain a better format of our responses (Figures were not well pasted here). Thank you for your understanding,
Véronique Fontaine
“ Several figure images are blurry. Please provide higher-resolution versions”.
Our response: We apologize. Figures with higher resolutions have been sent and are now present in the revised manuscript.
“Can the authors compute and provide detailed aerosolization parameters such as MMAD, geometric standard deviation (GSD), and respirable fraction? This information is essential and should be included in the results.”
Our response: The aerodynamic assessment of fine particles of the dry powders was conducted using the Fast Screening Impactor due to the sensitivity limitation of the analytical method, even when using ten capsules. With this limitation, only the fine particle dose (respirable dose) and fine particle fraction (respirable fraction) can be determined, not MMAD and GSD. Even though we believe also these data are very important, we apologize for this limitation and are unable to provide MMAD and GSD.
“From the drug release studies, it appears that 80–95% of VAN and THL are released within 10 minutes. Is this release window sufficient for uptake by alveolar macrophages? Are the particles sufficiently solid and stable for intracellular targeting to infected macrophages or the lung interstitium?”
Our response: we thank the reviewer for this question. The dissolution test is performed to discriminate the formulation rather than to simulate dissolution or release in the lung as mentioned clearly from line 703 to line 711.
“It should be noted that the dissolution medium used in this work was not physiological (i.e., presence of 3% SDS in PBS) but was necessary to allow determination of both drugs by HPLC-DAD method. Therefore, the presence of 3 % SDS increase the dissolution rate of THL and VAN from the microparticles compared to physiological conditions encountered in the lungs. THL is more difficult to solubilize under physiological conditions and slows down the release of VAN release before its complete dissolution. This could potentially favour AM phagocytosis of the undissolved parts of the formulation, while the dissolved parts could act either by diffusing through the AM cell membrane or by acting on extracellular Mtb. “
Therefore, sink conditions are required and sufficient quantification by the analytical methods during the test. The dissolution medium was optimized based on these criteria, and 3% of sodium dodecyl sulfate was added. Although this is not physiological, it allows for quantifiable THL solubilization while maintaining sink conditions.
“The statistical analysis in Figure 3A should be carefully rechecked. The error bars, especially for the VAN/THL/50% HCO group, do not seem to support the claimed statistical significance”.
Our response: We verified the statistical analyses for this Figure and it was indeed statistically significant as you can see:
Crude data:
“Why were different strains of Mycobacterium used for the in vitro and in vivo studies? This discrepancy affects the translational value of the data. The authors may consider either presenting in vitro data using H37Rv or omitting M. bovis data altogether to ensure consistency.”
Our response: We understand this comment, however, we were unable to perform all in vitro study in BSL3 with M. tuberculosis. Indeed, although we were able to perform certain experiments in the BSL3 of Pasteur Institute in Lille, before 2018 (PhD student Celine Rens’experiments with Rifampicin), for security reasons, PhD student Faustine Ravon was not allowed in 2018 to work in the same BSL3). However, in vitro experiments were intended to select for the best formulation before in vivo experiments. We therefore used M. bovis BCG::ESX-1Mmar, producing PDIM as all tuberculous mycobacteria, but also known to have a similar or even lower susceptibility to vancomycin with THL (Rens et al, 2016; doi: 10.1128/AAC.00872-16.), being a worse case scenario inoculum to assess antimycobacterial activity. Furthermore, we used for the first time a new vaccine M. bovis BCG genetically modified strain (expressing ESX-1 from M. marinum), so that the mycobacteria would also be localized in the cytosol, as M. tuberculosis. This new in vitro model would be an interesting and innovative model for many laboratories working in BSL2.
Why was RFU (Relative Fluorescence Units) chosen as the primary readout for bacterial burden instead of the standard CFU plating method? RFU data tends to be highly variable and may not reliably distinguish between bacteriostatic and bactericidal effects.
Our response: Indeed, we intended to have the CFU data but due to experimental errors, during dilutions, or media preparation, CFU data could not be used. We then used only the RLU.
The manuscript’s readability could be improved by moving non-essential to the supplementary section and streamlining the Results and Discussion sections or more appropriately as separate “Discussions” section. The results are too mixed and there is not a proper story. The appendix section makes it more confusing.
Our response: We agree and try to simplify the take home messages in our text. We also removed one redundant chapter (present in the Material and methods and in the Appendix) and we moved the Appendix in the Supplementary Information as well as the purchase information.
We thank you for your advices and hope that our manuscript is now acceptable for publication in “Pharmaceutics”.
Yours sincerely,
Véronique Fontaine

Reviewer 2 Report
Comments and Suggestions for Authors
In the manuscript “Assessment of Innovative Dry Powders for Inhalation of a Synergistic Combination against Mycobacterium tuberculosis in infected Macrophages and Mice” the authors aimed to develop and assess the efficacy of dry powders for inhalation of VAN microparticles embedded with THL. The authors reports that HCO enables the co-release of VAN and THL in aqueous media at the desired weight ratio. Microparticles with 0% to 50% HCO drastically reduced cytoplasmic Mycobacterium bovis survival, with higher efficacy (99.9% to 62.5 33 %) at low HCO concentration.
Although the study reports some good findings but the combination of vancomycin (VAN) and tetrahydrolipstatin (THL) which has shown effective synergistic action in vitro against Mycobacterium tuberculosis (Mtb) is not clear. The VAN and THL concentration and treatment time is not clear.
The manuscript has many grammatical, technical and scientific errors in presentation. Discussion part has been poorly described. Its should be in depth with possible reason of results. Conclusion is also poor and need to elaborate with main findings in the current study.
Abstract
Lines 20-21 should be revised.
Line 32 [desired weight ratio.] what does it mean?
Line 33, % should be from low to high.
Line 37 [significant bacterial burden reduction] need revision.
Line 36-367 should clearly provide the concentration and days of treatment and reduction.
Conclusion should rewrite with exact concentration and duration of treatment.
Introduction
Lines 45-46 also need revision
Line 55 [causative agent of pulmonary TB]? Mtb is also causative agent of extra pulmonary TB.
Line 86 [aerodynamic diameter (dae) of the drug-based formulation should be between 1 and 5 µm] provide a reference/cite it.
Material and Methods
Line 158 to 178, the purchase information may be provided as supplementary information,
Line 311-312 need revision
The methodology is very long and contain some information as supplementary.
The methodology section may also be in flowchart for better understanding the flow and steps.
Which guidelines were followed for animal protocol?
Line 383 [50 rpm twice for 15 minutes]. It better to use RCF as the diameter is different in different centrifuge.
Lines 398 [anaesthesia by ketamine and xylazine at 150 mg/kg and 2 mg/kg, respectively]. Ketamine for mice is commonly used at 100–150 mg/kg — your 150 mg/kg dose is at the upper end, but still acceptable. 2 mg/kg seems very low. The authors may check it again.
How many mice have been used in treatment/trial should be provided in table form for better understanding?
Results
Figure 1, “An ANOVA was applied with Bonferroni’s post-hoc tests” This may be removed. Already mentioned in methods.
Line 598, “Fine particle” should be lower case letter.
Line 620 remove ANOVA
Lines 626-28 need revision
Figure 3, The table should be separated.
The ANOVA should be removed from figures captions.
Lines 706-07 need revision
Line 802 need revision
Line 814, significant weight loss? Value?
Conclusion should be revised.
References
Some more updated references may be cited.
Author Response
Dear reviewer 2,
Thank you for your advices concerning our manuscript entitled “Assessment of Innovative Dry Powders for Inhalation of a Synergistic Combination against Mycobacterium tuberculosis in infected Macrophages and Mice”. Please find hereunder our point-by-point responses to the issues raised by the reviewers, with references to line numbers of the revised file version with track changes:
Reviewer 2 asked for:
“Although the study reports some good findings but the combination of vancomycin (VAN) and tetrahydrolipstatin (THL) which has shown effective synergistic action in vitro against Mycobacterium tuberculosis (Mtb) is not clear. The VAN and THL concentration and treatment time is not clear.
The manuscript has many grammatical, technical and scientific errors in presentation. Discussion part has been poorly described. Its should be in depth with possible reason of results. Conclusion is also poor and need to elaborate with main findings in the current study.”
Our response: see detailed answers below.
Abstract
“Lines 20-21 should be revised.”
Our response: This is now done, see lines 20-22.
“Line 32 [desired weight ratio.] what does it mean?”
Our response: Indeed, this is not appropriate, and as we already said “co-release”, this part was removed in the abstract, see lines 32-33.
“Line 33, % should be from low to high.”
Our response: This can’t be changed because it is for 0 and 50% HCO, respectively. This is now better presented line 34-37.
“Line 37 [significant bacterial burden reduction] need revision.”
Our response: Statistic analyses have been performed and verified. Please see enclosed our crude data analysis in the responses to Reviewer 1. We change this sentence, lines 38-42 in the abstract to give more accurate information.
“Line 36-367 should clearly provide the concentration and days of treatment and reduction”.
Our response: Thank you for this suggestion. This is now done lines 38-42.
“Conclusion should rewrite with exact concentration and duration of treatment.”
Our response: Thank you for this suggestion. This is now done lines 44-45.
“Introduction
Lines 45-46 also need revision”
Our response: Thank you for this suggestion. This is now done line 52-53.
“Line 55 [causative agent of pulmonary TB]? Mtb is also causative agent of extra pulmonary TB.”
Our response: Indeed, thank you for this remark. This is now removed line 64.
“Line 86 [aerodynamic diameter (dae) of the drug-based formulation should be between 1 and 5 µm] provide a reference/cite it.”
Our response: This is now done, line 97.
“Material and Methods
Line 158 to 178, the purchase information may be provided as supplementary information",
Our response: Indeed, this is now in the supplementary information.
“Line 311-312 need revision”
Our response: This is now revised, see lines 331-332.
“The methodology is very long and contain some information as supplementary”.
Our response: Indeed, we removed or summarized some sentences (see lines 238, 281-285, 286-288, 307-311, 326-331, 371, 410-420, 469-470, 473-474, 499-500), one redundant chapter (see lines 502-530) or moved some sentences to supplementary information (The purchase information (lines 167-189), as suggested)
“The methodology section may also be in flowchart for better understanding the flow and steps.”
Our response: As we improve the text, we hope now that the take home message is clearer and that the manuscript text follows logical steps.
“Which guidelines were followed for animal protocol?”
Our response: a previous tolerance study on healthy mice for the VAN and THL a the used dosage showed no signs of toxicity. During the efficacy study on infected mice, no signs of toxicity could be attributed to the drugs, only to the disease. When we added oral rifampicin to this treatment at a dose safe for this strain of mice, weight loss until human endpoint was observed after 1 week, whereas no signs of toxicity were seen in the oral rifampicin-treated group. Therefore, we conducted further investigation to understand the reasons for this by performing a tolerance study using rifampicin at different dosages and the VAN/THL/25% HCO at 500 mg/kg VAN and 50 mg/kg THL. During this experiment, we observed that the weight loss was due to the cumulative techniques of oral gavage and endotracheal administration under general anesthesia. This issue will be addressed by adding Solid drink supplementation for the mice.
This information is already available in the text and is described in details in the supplementary data (section 6)
“The first experiment was planned for 2 weeks but lasted 1 week due to excessive weight loss until the humane endpoint (20% of initial weight) due to lower food and water intake of mice under cumulative oral and endotracheal administrations (supplementary data). ….As human endpoints were reached, further investigations were done before the second experiment to assess the reasons of weight loss (supplementary data).
The second experiment was planned for 2 weeks and lasted 2 weeks without reaching human endpoints thanks to the supplementation of Solid Drink® allowing to overcome the negative cumulative effect of oral and endotracheal administrations on body weight. “ (lines 485-488)
“Line 383 [50 rpm twice for 15 minutes]. It better to use RCF as the diameter is different in different centrifuge”.
Our response: This was performed in a mixer (three-dimensional Powder Blender Mixer Turbula® (Willy A. Bachofen AG, Muttenz, Switzerland). We were only able to set up based on the rpm parameter.
“Lines 398 [anaesthesia by ketamine and xylazine at 150 mg/kg and 2 mg/kg, respectively]. Ketamine for mice is commonly used at 100–150 mg/kg — your 150 mg/kg dose is at the upper end, but still acceptable. 2 mg/kg seems very low. The authors may check it again”.
Our response: this mixture composition was advised by our veterinary and was used without issues.
“How many mice have been used in treatment/trial should be provided in table form for better understanding?”
Our response: Please find hereunder the amount of mice participating in our study. This information is now available in supplementary information (section 5.1.).
|
VAN/THL or VAN/THL/25% HCO based experiments |
||
|
Experiment 1 |
Number of mice/group |
Treatment duration |
|
CRT |
8 |
2 weeks |
|
VAN/THL |
5 |
2 weeks |
|
Experiment 2 |
Number of mice/group |
Treatment duration |
|
CRT |
14 |
|
|
HCO |
16 |
2 weeks |
|
VAN/THL/25% HCO |
20 |
2 weeks |
|
Rifampicin-based experiments |
||
|
Experiment 1 |
Number of mice/group |
Treatment duration |
|
RIF 10 mg/kg |
12 |
1 week |
|
RIF10mg/kg + VAN/THL/25% HCO |
12 |
1 week |
|
Experiment 2 |
Number of mice/group |
Treatment duration |
|
RIF 7.5 mg/kg |
10 |
2 weeks |
|
RIF 7.5 mg/kg+ VAN/THL |
10 |
2 weeks |
“Results
Figure 1, “An ANOVA was applied with Bonferroni’s post-hoc tests” This may be removed. Already mentioned in methods.”
Our response: Indeed, this is now done in all figure legends, lines 650, 747, 805, 884.
“Line 598, “Fine particle” should be lower case letter.”
Our response: Indeed, this is now done.
“Line 620 remove ANOVA”
Our response: This is now done, line 665.
“Lines 626-28 need revision”
Our response: This sentence was removed, line 679-680.
“Figure 3, The table should be separated.”
Our response: We removed the Table in Figure 3 and wrote the data in the text.
“The ANOVA should be removed from figures captions”.
Our response: Thank you for this suggestion. This sentence was removed from all figure legends, see lines lines 650, 747, 805, 884.
“Lines 706-07 need revision”
Our response: This is now done, line 763-767.
“Line 802 need revision”
Our response: This is sentence has been removed, as it was redundant, see line 874-876.
“Line 814, significant weight loss? Value?”
Our response: Loss of 20% or more weight loss was considered as significant. This is now added, line 888-889.
“Conclusion should be revised.”
Our response: This is now done, lines 922-931.
“References
Some more updated references may be cited.”
Our response: Reference 4 (2017) was replaced by Raman SK, Roy T, Verma K, Yadav C, Verma S, Deivreddy VSR, Sofi HS, Bharti R, Sharma R, Bansode H, Kumar A, Sharma RK, Singh J, Mugale MN, Bajpai U, Jain V, Singh AK, Misra A. Dry powder Inhalation of lytic mycobacteriophages for adjunct therapy in a mouse model of infection with Mycobacteriumtuberculosis. Tuberculosis (Edinb). 2025 May;152:102631. doi: 10.1016/j.tube.2025.102631. Epub 2025 Mar 11. PMID: 40088506.
Reference 2 (2020) was replaced by Omoteso OA, Fadaka AO, Walker RB, Khamanga SM. Innovative Strategies for Combating Multidrug-Resistant Tuberculosis: Advances in Drug Delivery Systems and Treatment. Microorganisms. 2025 Mar 24;13(4):722. doi: 10.3390/microorganisms13040722. PMID: 40284559; PMCID: PMC12029526.
We thank you for your advices and hope that our manuscript is now acceptable for publication in “Pharmaceutics”.
Yours sincerely,
Véronique Fontaine

Round 2
Reviewer 1 Report
Comments and Suggestions for Authors
Dear Authors,
Thank you for addressing the queries. The rebuttals to most of the queries are appropriate and have been clearly articulated. However, I would still recommend that the authors include a distinct "Discussion" section in the revised manuscript, as this would enhance the clarity and flow of the manuscript for the benefit of readers.
Regards
Author Response
Dear reviewer,
Thank you for your nice review. We would like to asked you whether you could accept that we don't separate the results and the discussion. We know our manuscript is long but, thanks to changes introduced after reviewing, we believe we improved the take home message of this manuscript and the different steps of this study is now easier to follow. Furthermore, the first author is not anymore in the lab, so, to rewrote 50% of the manuscript in 3 days, doesn't sound for us as a minor change. We thank you in advance for your understanding and again for helping us to improve ur manuscript. Sincerely yours, Véronique Fontaine
Reviewer 2 Report
Comments and Suggestions for Authors
The authors successfully adressed all my previous comments.
Author Response
Dear reviewer,
Thank you very much for your help and nice review. Sincerely yours, Véronique Fontaine